# ICP-MS Determination of Antimicrobial Metals in Microcapsules

**DOI:** 10.3390/molecules27103219

**Published:** 2022-05-18

**Authors:** Iva Rezić, Maja Somogyi Škoc, Mislav Majdak, Slaven Jurić, Katarina Sopko Stracenski, Kristina Vlahoviček-Kahlina, Marko Vinceković

**Affiliations:** 1Department of Applied Chemistry, Faculty of Textile Technology, University of Zagreb, 10000 Zagreb, Croatia; mislav.majdak@ttf.hr; 2Department of Material Testing, Faculty of Textile Technology, University of Zagreb, 10000 Zagreb, Croatia; maja.somogyi@ttf.hr; 3Faculty of Agriculture, University of Zagreb, 10000 Zagreb, Croatia; sjuric@agr.hr (S.J.); ksopko@agr.hr (K.S.S.); kvkahlina@agr.hr (K.V.-K.); mvincekovic@agr.hr (M.V.)

**Keywords:** ICP-MS, microcapsules, microwave digestion, silver, zinc

## Abstract

Silver (Ag) and zinc (Zn) are very powerful antimicrobial metals. Therefore, in this research, a high-throughput, sensitive, and rapid method was developed for the determination of Ag and Zn in microcapsules using inductively coupled plasma mass spectrometry (ICP-MS). The sample preparation procedure employed simple microwave digestion of the microcapsules with 55.55% *v*/*v* HNO_3_ and 44.45% *v*/*v* H_2_O_2_. The method was applied to determine Ag and Zn in microcapsule samples of different sizes (120 and 450 μm) after their preparation with and without chitosan. Prepared microcapsules, after characterization, were bonded to a polymer carrier by sol-gel procedure and the materials were characterized by FTIR spectroscopy and high-resolution optical microscopy. Significant differences were found in Ag and Zn levels between microcapsules samples prepared with and without chitosan. The results have shown that samples with chitosan had up to 20% higher levels of Zn than Ag: 120 μm microcapsules contained 351.50 μg/g of Ag and 85.51 μg/g of Zn, respectively. In contrast, samples prepared without chitosan showed larger overall variability: In microcapsules with a diameter of 120 μm, the amounts of antimicrobial metals were 98.32 μg/g of Ag and 106.75 μg of Zn, respectively. Moreover, 450 μm microcapsules contained 190.98 μg/g of Ag and 121.35 μg/g of Zn. Those quantities are high enough for efficient antimicrobial activity of newly prepared microcapsules, enabling the application of microcapsules in different antimicrobial coatings.

## 1. Introduction

Due to the increase in antimicrobial resistance to conventional pharmaceutical drugs, the World Health Organization has recognized this problem as one of the most important issues in world health protection. One of the recent solutions is silver that is very well known as an efficient antimicrobial reagent [1]. Moreover, silver nanoparticles show strong antimicrobial activity not only on bacteria but as well as on spores resistant to high temperature and high pressure [2]. In addition, microcapsules that can be filled with silver ions or silver nanoparticles are easily combined with polymer carriers to obtain efficient antibacterial materials.

The microcapsules originated from the 1940s when the pharmaceutical and paper industries began introducing encapsulated products. The process of encapsulation contains several steps in which the active compound is surrounded by a natural, semi-synthetic or synthetic polymer shell coating, with a typical diameter between 2 and 2000 μm. The thickness of the walls surrounding the active species is usually 0.5–150 μm, with an average proportion of core between 20 and 95% *w*/*w*. [3]. Over the past 10 years, the microcapsules have been extensively used in different sectors, from agricultural, pharmaceutical, and cosmetic to textile industrial products. In addition, the total number of commercial microcapsule applications has grown, encouraging the use of silver for different applications [3]. There are many reasons why the microencapsulation achieves many benefits, but the two main facts are blocking the active substances within the walls of microcapsules, and gradual release of the active ingredient under the desired environmental conditions.

The encapsulation process can vary under different physical and chemical methodologies, and the main approaches are physical methods of encapsulation: air suspension coating, coextrusion, rotating disk atomization, spray-drying or cooling, fluid coating, and centrifugal extrusion; or chemical methods of encapsulation: coacervation, evaporation of the solvent, in-situ polymerization, liposome technology, matrix polymerization, nanoencapsulation, and solvent extraction.

The choice of a particular technique will define the final outcome results: the usage, inertness, release time, desired concentration of the active species, the mechanism of release, particle size, density and stability, and also the price of the final product. There are many advantages of microencapsulation, but the main one is protecting the sensitive active species with the controlled and targeted release, and prevention of degradation reactions such as oxidation or dehydration. The graphical presentation of the structure of the microcapsules is shown in Figure 1.

To analyze small microencapsulated products, the precise and accurate analytical methods such as inductively coupled plasma mass spectrometry (ICP-OES), inductively coupled plasma optical emission spectrometry (ICP-OES), time of flight secondary ion mass spectrometry (ToF-SIMS), scanning electronic microscope (SEM), and others are the most appreciated [4,5]. In samples such as microencapsulated active silver species, very small changes in the concentration of encapsulating components can have a significant impact on the property of materials. Therefore, it is crucial to have a high degree of precision while measuring the trace and minor components in the presence of other macro compounds in microcapsules.

In cases when sensitive compounds are encapsulated, the problems could arise from the fact that valuable silver materials are available for analysis in very small sample amounts. Secondly, only accurate analytical methods may provide information on the concentration of silver nanoparticles or silver ions that are released from the microcapsule during prolonged contact with different environmental conditions [6,7,8,9]. For simultaneous determination of many elements present in the same microcapsules, ICP-MS and ICP-OES are much more appropriate techniques than analytical absorption spectrometry (AAS) or graphite furnace AAS (GF-AAS), since those are single-element methods for metal analysis demanding longer time for the analysis [10,11,12,13].

Secondly, microscopic methods such as AFM, SEM, or high resolution OM can be applied as non-destructive methods for the characterization of microcapsules [14,15,16,17]. However, the drawback of the SEM-EDS chemical analysis is that the obtained results present the chemical composition of the sample surface and not the whole sample [18,19]. Therefore, in cases with hollow microcapsules, the real chemical composition cannot be obtained without the combination with ICP-MS or ICP-OES [20,21,22,23,24,25,26]. Some authors have recommended a chemometric approach to discriminate silver ions and nanoparticles in their binary mixtures of consumer products by graphite furnace atomic absorption spectrometry [13]. Others propose an ultrasonic extraction, an electron probe micro-analysis using a set of wavelength dispersive spectrometer, or Resonance Rayleigh scattering [27,28,29,30,31,32]. The drawback of such methodologies is the limited availability and accessibility of such equipment. Characterization of the layer structure can be performed by Auger electron spectroscopy with the inconvenience of demanding ultra-high vacuum [33]. Moreover, an external differential PIXE technique can be applied as a non-destructive characterization methodology that also allows for the detection of the layer structure, which is non-destructive for samples [34].

During complex quantification of silver nanoparticles in the presence of silver ions, proper quality control procedures have to be applied to detect and characterize the samples regarding their size and concentration. Laborda et al. showed that the unknown nature of the nanoparticles and small sizes with instrumental constraints limited the information achievable in the analysis of different consumer products, foods, and environmental or biological samples [35]. Wang et al. reported methodology for the speciation analysis of silver nanoparticles and silver ions Ag^+^ based on the separation techniques coupled with atomic spectroscopy, including extraction method, chromatographic method, cation exchange reaction (CER), and single particles (SP) detection [36]. They showed that there is a growing usage of products with silver nanoparticles that are released into the environment, as well as that silver ions can be detected in the environment after exposure to nanoparticles. The straightforward procedure contains the methodology for the determination of the total metal content and usually employs the microwave digestion as a sample preparation step, prior to the quantification by ICP-OES or ICP-MS [37,38,39]. In cases when alkali or earth alkali metals are elements of interest, absorption spectrometry is recommended [40,41], but for multi-elemental analysis, only the ICP-OES and ICP-MS can offer fast, routine, high-throughput results [41,42].

Trace element analysis is crucial for samples in which very small amounts of metals provide substantive effects on different strains of microorganisms [43,44,45]. In our previous work, we showed that metals such as zinc and silver can have very strong antimicrobial effects, which is applicable in many investigations, especially when they are used in the form of nanoparticles [46,47,48,49,50,51]. Therefore, the goal of this research was to quantitatively determine the amounts of antimicrobial metals in microcapsules prepared with and without chitosan by the combination of different methods. Since the efficiency of antimicrobial microcapsules is defined by the compatibility of the core and shell materials, this work was focused on the characterization of microencapsulated antimicrobial silver inside the zinc sulfate shells, encapsulated for application on medical textiles.

## 2. Results and Discussion

Since only accurate analysis can provide information on the chemical composition of microcapsules, a very important part of the analysis was the determination of the limit of detection (LODs). In this work, limits of detection for all elements of interest were expressed as three times the standard deviation of the blank solution and reached the following values: 107 μg/L for silver and 66 μg/L for zinc, respectively. Table 1 shows the ICP-MS results of elemental composition of investigated samples obtained after microwave digestion.

As it can be seen from Table 1, the method was applied to determine Ag and Zn in microcapsules samples of different sizes (120 and 450 μm) after their preparation with and without chitosan. Significant differences were found in Ag and Zn levels between microcapsules samples prepared with and without chitosan. Firstly, while Zn levels were up to 20% higher in the group of samples prepared without chitosan, compared with the group with it, the Ag content showed larger overall variability (more than 72%).

Unfortunately, the proposed methodology is destructive and demands a prior degradation of all samples. However, the advantages of ICP-MS are high throughput as well as fast and precise analysis of trace amounts of samples. Secondly, limits of detection in ICP-MS are much lower than in ICP-OES, which is extremely important in cases of samples in which minor and trace elements have to be determined in very small sample amounts available for the analysis. Therefore, the best choice is to perform fast testing by ICP-MS and then to monitor the leaching of particular silver ions by an independent method that provides speciation. In cases of more complicated samples that contain traces of analyte, the better choice would be GF-AAS, a method for single element analysis, but with limits of detection lower than in AAS and ICP-OES. Encapsulation was performed as previously described, and the samples were recorded under high-resolution optical microscope. The resulting microphotographs are presented in Figure 2.

As can be seen from Figure 2, microencapsulation enables hollow spheres filled with antimicrobial silver that can be released in desired time intervals. The range of potential applications of such microencapsulation in different products is huge and covers different pharmaceutical, cosmetic, textile, agricultural, and paper industries. Since materials for forming microcapsules are mainly carbohydrates (starch, cellulose, and gums), proteins (milk proteins—caseins and whey proteins, gluten, and gelatin) and lipids (fatty acid-line, alcohols, glycerides, waxes, and phospholipids) or a combination thereof, microcapsules are compatible with human skin and are therefore becoming increasingly interesting for the functionalization of textile polymers.

Therefore, as a potential application in producing medical textiles, the microcapsules were in this investigation fixed to the textile carrier by sol-gel methodology and dip coating techniques, and the resulting material is presented in Figure 3. The FTIR spectra of the material is presented in Figure 4. The antimicrobial activity of silver nanoparticles was positively tested in our previous investigation on *metil resistant Staphylococcus aureus* (MRSA) and *metil sensitive Staphylococcus aureus* (MSSA) strains of microorganisms [43]. In addition, silver ions and zinc are also very strong antimicrobial agents [44,45].

Therefore microcapsules might enable a very efficient solution for application in hospitals. Furthermore, microencapsulation can be an indispensable part of the functionalization of other polymer materials, such are for example agrotextiles or geotextiles. Currently, although microcapsules can stay un-attacked during 25 to 30 washing cycles, processes such as ironing and drying can cause a dramatic reduction of the desired effects [3]. Nevertheless, microencapsulation provides long-term results for certain goals, e.g., to deliver biologically active substances to desired places. Silver ions and nanoparticles having anti-bacterial effects on a wide range of Gram-positive and Gram-negative bacteria, including antibiotic-resistant strains, are therefore widely used.

As can be seen from the Figure 5, SEM and SEM-EDX results enabled determination of surface morphology, determination of the microcapsules dimension, as well as the chemical composition of the surface of the microcapsules. The antimicrobial elements detected by SEM-EDX were Ag and Zn, in 0.09 and 4.65 mass percentages, respectively. A similar methodological approach was applied by Xu et al. [46], who characterized the morphology and composition of the Fe_3_O_4_/CS-Ag microcapsules by Fourier transform infrared (FT-IR) spectra, scanning electron microscopy (SEM), energy dispersive X-ray spectroscopy (EDX), and X-ray photoelectron spectroscopy [46]. In addition, Neri et al. used SEM imaging and EDX elemental mapping for characterization of Si, O, and Ag arising from microcapsules [47]. Their research was focused on prepared silver (Ag)-grafted PMA (poly-methacrylic acid) capsules loaded with sorafenib tosylate (SFT) as an anticancer drug that allowed on-demand control of the dose, timing, and duration of the drug release. In contrast, our research was focused on Ag and Zn as antimicrobial substances and on the determination of their amounts in antimicrobial microcapsules. However, SEM and SEM-EDX were very favorable techniques for the characterization of different microcapsules, as was described in the aforementioned cases.

In addition, we applied ICP-MS to determine the total Ag and Zn content of the samples to predict their full antimicrobial effects. ICP-MS is highly recommended as an excellent choice for the multi-elemental determination of metals in sample solutions with high precision and accuracy. It offered the possibility of determining the trace, minor and major elements in the same sample. SEMEDX analysis enabled detecting the microcapsules dimensions, distribution of Ag and Zn in the shell, and homogeneity of different components. The combination of both methods enables the complete characterization of antimicrobial microcapsules. Figure 6 and Figure 7 show the results of the ICP-MS determination. Figure 6 is a graphical comparison of concentration of antimicrobial metals in microcapsules prepared with and without the chitosan, of 120 μm in size, and Figure 7 presents the ICP-MS spectra of all samples.

Figure 6 clearly presents the difference among samples prepared with and without chitosan. The difference in the amounts of antimicrobial silver is significantly higher in samples prepared with chitosan (351.50 μg/g of Ag), than in samples that were prepared without it (98.32 μg/g of Ag).

In contrast, the levels of zinc are a little bit lower in samples prepared with chitosan (85.51 μg/g of Zn compared to 106.75 μg of Zn), which was expected since both the chitosan and zinc are a part of the shell of the microcapsules and not the compounds in their core.

All ICP-MS results are presented in Table 1 and show that samples with chitosan had 20% higher levels of Zn than Ag. Precisely, microcapsules with a diameter of 120 μm contained 351.50 μg/g of Ag and 85.51 μg/g of Zn, respectively. The opposite case was observed with samples prepared without chitosan. Those samples showed much larger overall variability. The microcapsules with diameters of 120 μm contained lower amounts of antimicrobial metals: 98.32 μg/g of Ag and 106.75 μg of Zn, respectively. Much bigger microcapsules prepared with an average diameter of 450 μm, contained 190.98 μg/g of Ag and 121.35 μg/g of Zn. So, when we compare the obtained results, it can be concluded that 3.75 times bigger microcapsules contained only two times more silver. All of those quantities are high enough for efficient antimicrobial activity of newly prepared microcapsules, enabling their application on different antimicrobial materials.

The efficiency of the quantitative determination of the trace element (metal) amounts in samples strongly depends on the proper sample preparation. We applied the microwave digestion by mixing 2.5 mL of spectral pure nitric acid and 2.0 mL of hydrogen peroxide. The digestion was performed in one cycle with three steps until maximal temperature of 170 °C was achieved. This protocol is based on our previous research and is similar to the methodology of microwave digestion reported by Khan et al. [52]. They optimized the methodology of microwave digestion by using reagents nitric acid (HNO_3_, 65%, 5 mL) and hydrofluoric acid (HF, 40%, 2 mL). Their obtained results revealed excellent performance with recovery values of metal ions ranging between 99.33% and 105.67%.

ICP-MS spectra are shown in Figure 7. They show spectral data used for used for the determination of the concentration of metal ions. Three parts of Figure 7 present three different results obtained for three different samples of microcapsules filled with silver, which were investigated after their digestion in the microwave oven. Firstly, 120 μm microcapsules prepared without chitosan are shown in Figure 7A. Secondly, Figure 7B presents the data of 450 μm microcapsules prepared without the chitosan, while Figure 7C presents the data of 120 μm microcapsules prepared with the chitosan.

As can be seen from the Figure 7, and confirmed in the literature, silver occurs naturally as a mixture of two stable isotopes: 107Ag (51.8%) and 109Ag (48.2%). Szymanska-Chargot et al. [53] reported that the difference in silver ICP-MS spectrum is the consequence of the stable silver isotopes: Firstly, the peak corresponding to the Ag^+^ ion consists of two peaks that are corresponding to the isotopes of silver. In addition, silver oxide as Ag2O^+^ can have different combinations of Ag and O atoms: 107Ag2O^+^, 107Ag, 109AgO^+^, and 109Ag2O^+^ [53]. Moreover, the 107Ag is causes polyatomic interference with 91Zr16O^+^ and 109Ag to 92Zr16O1H^+^. Therefore, in complex systems, this should be checked. However, in our case, the spectrum was free of all interferences. In addition, the zinc isotope ratios are as follows: ^64^Zn^+^/^66^Zn^+^, ^67^Zn^+^/^66^Zn^+^, ^68^Zn^+^/^66^Zn^+^, and ^70^Zn^+^/^66^Zn^+^. Zn oxides are found at 80 (64Zn16O^+^) and 82 (66Zn16O^+^), and ArZn at 104. In addition, ZnCl mass were found in the range of 99–105, while the isotopic peaks of ZnCl_3_^−^ were at 168.84, 170.83, 171.83, 172.83, 173.82, 174.82, 175.82, 176.82, 177.82, and 178.82 *m*/*z.*

The size of microcapsules has a crucial role in their application and in vivo efficiency. Wang and Hu reported that microparticles larger than 100 nm can stay at the site of administration. They investigated the processes of lymphatic uptake and accumulation and found them to be the most significant between 10–80 nm [48]. Therefore, the determination of the morphology of microcapsules and nanocapsules is very important. We applied SEM and SEM-EDX investigation, which is an addition to the other methodologies that can be used for the same purpose, such as AFM, laser light diffraction (LD) method, or Coulter counter, as was reported by Lengyel et al. [49]. They showed that the particle size analysis of microparticles that are bigger than 3 μm is in most cases carried out based on laser light diffraction (LD) method or using a Coulter counter. They emphasized that in the LD methodology, the calculated distribution indicates the span-value as a measure of the size distribution. In addition, they listed a polydispersity index (PDI) that is determined by dynamic laser light scattering as an indicator of the size distribution in the lower size region of microparticles [49].

Shekunov et al. recommend the AFM as a technique that provides information about the surface of microparticles by profiling the surface [50], while Shahi [51] proposed a Coulter counter that gives an absolute particle number per volume unit for various size ranges of microparticles. For this reason, such methodology is important in the particle analysis of microparticles for intravenous usage [51].

We successfully applied SEM, SEM-EDX, FTIR, and ICP-MS techniques for the characterization of microcapsules. From the obtained results we conclude that the range of the Ag and Zn metals was in correlation with the current limits necessary for antimicrobial effects according to the literature data, and this is the basis of our future steps in the preparation of powerful antimicrobial coatings for medical materials. In our further experiments, we plan to perform experiments with the monitoring of leaching quantities of silver and its release as active antimicrobial species.

## 3. Materials and Methods

### 3.1. Instrumentation

An inductively coupled plasma mass spectrometer, Agilent 7900 ICP-MS (Agilent Technologies, Singapore), was used for all measurements. The instrument was equipped with standard nickel sampling and skimmer cones, a standard glass concentric nebulizer, quartz spray chamber, and a quartz torch with a 2.5 mm id injector. Operating conditions were as follows: sample depth 10 mm; ORS mode NO GAS/He; nebulizer carrier gas flow rate 1.09 L/min; extract lens 1 was zero, and 2 was −190; RF power 1550 W; RF matching 1.80 W; spectral detection mode; and integration time 0.1 s. All the measurements were done in triplicates.

### 3.2. Reagents and Solutions

All the reagents (nitric acid), silver nitrate, and silver nanoparticles (colloidal stabled suspension), and 1000 μg/mL standards (Ag and Zn) used for this research work were of p.a. grade, supplied by Merck, Darmstadt, Germany. Ultra-pure nitric acid supplied by Merck, Darmstadt, Germany, and ultrapure water produced by NIROSTA (Millipore, Burlington, MA, USA) were used in ICPMS analysis. Textile samples used for functionalization were pure cotton and viscose materials intended for medical purposes.

### 3.3. Microencapsulation

Microcapsule samples were prepared as sodium alginate samples that contained silver in their core. In contrast, the outer shell was made of zinc sulfate 7-hydrate, with and without the chitosan. Encapsulation was performed by the reaction of sodium alginate and calcium chloride that produced calcium alginate and sodium chloride, according to the reaction:2NaC_6_H_7_O_6_ + CaCl_2_ → 2NaCl + C_12_H_14_CaO_12_(1)

After 30 min, the spheres were filtered, rinsed, and stored in a cool and dark place prior to the characterization, because otherwise they would disintegrate.

### 3.4. Microwave Digestion

A microwave digestion oven (Berghof, Darmstadt, Germany) was used to digest 25 mg of the microcapsule samples by 2.5 mL spectral pure nitric acid and 2.0 mL of hydrogen peroxide. The digestion was performed in one cycle with three steps until maximal 170 °C. After the digestion, the sample solutions without residues were diluted with ultrapure water and measured on ICP-MS. In addition, microcapsules with Au and Zn were, after preparation, analyzed by high-resolution OM microscopy.

### 3.5. Sol-Gel Procedure

Microcapsules with Ag and Zn were applied on woven viscose textile materials by dip coating (sol-gel) procedure using GLYMO precursor and HCl acid as a catalyst. After the immersion, the samples were dried for 24 h at room temperature. After drying, the samples were heated for 60 min at 100 °C to obtain a uniform coating suitable for antimicrobial protection (self-sterilizing or hygienic coatings). Several parameters that have main effects on the thickness of the covering film during immersion were optimized in our previous investigations: the speed of immersion, concentration and related viscosity of solutions, the surface tension, and the sample dimensions [43].

### 3.6. SEM-EDX Characterization

For SEM-EDX characterization of the microcapsules, a scanning electronic microscope TESCAN VEGA TS5136LS (manufacturer Tescan Vega, Brno, Czech Republic) equipped with an EDX detector was used.

## 4. Conclusions

The characterization of microcapsules filled with antimicrobial metals is an important part in the characterization of medical products. Interaction between the core and the shell of microcapsules can significantly influence the desired product properties. Therefore, the information on the exact chemical composition of trace and minor compounds is very important. Additionally, minor or trace compounds present in the core of microcapsules are released in time and can cause degradation under inappropriate cleaning and drying procedures, which will cause more degradation and collapse of microcapsules. Therefore, the chemical analysis and determination of chemical composition is a crucial step in preserving valuable active compounds. This work showed that inductively coupled plasma mass spectrometry and high-resolution optical microscopy may be easily applied for this purpose.

## Figures and Tables

**Figure 1 molecules-27-03219-f001:**
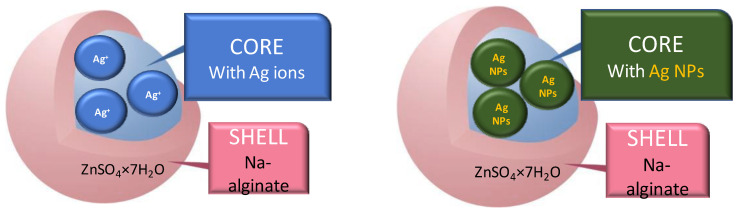
Graphical presentation of the structure of microcapsules that can be filled with silver ions or with silver nanoparticles, both having applications in cosmetic, pharmaceutical, and textile industries.

**Figure 2 molecules-27-03219-f002:**
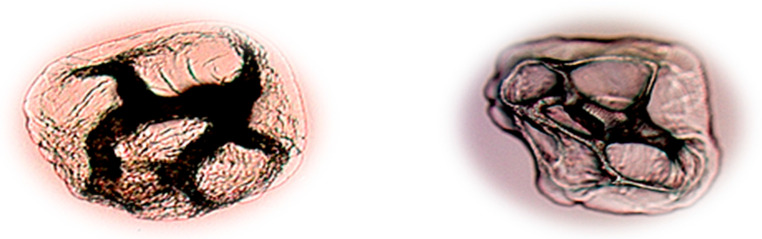
Microphotographs of microcapsules made by encapsulation with and without chitosan.

**Figure 3 molecules-27-03219-f003:**
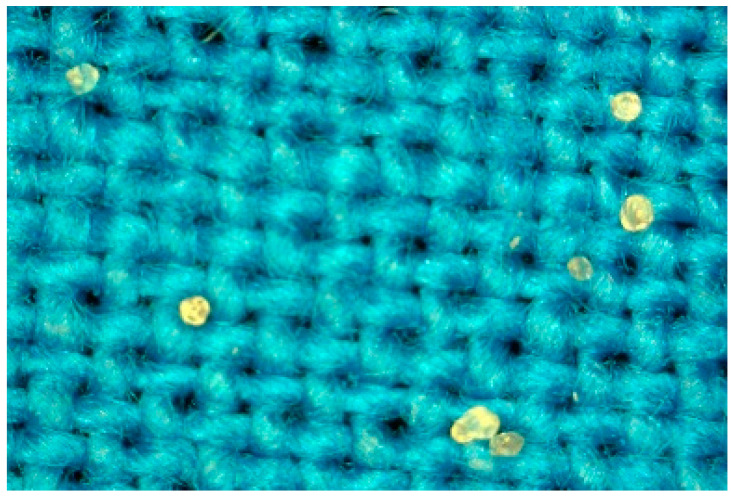
Textile material functionalized with microcapsules containing silver and zinc.

**Figure 4 molecules-27-03219-f004:**
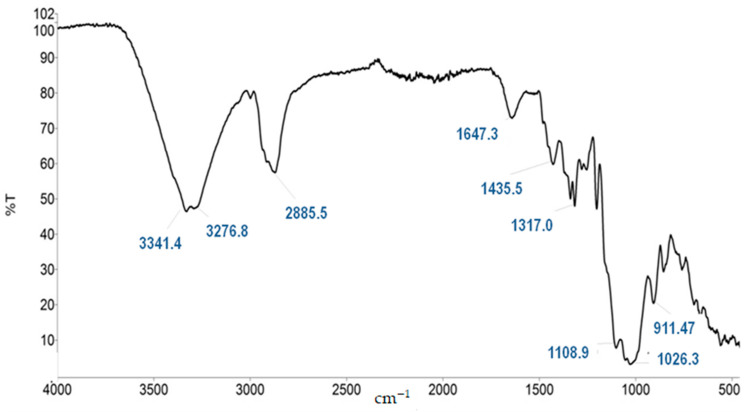
FTIR spectra of the textile material before functionalization with microcapsules.

**Figure 5 molecules-27-03219-f005:**
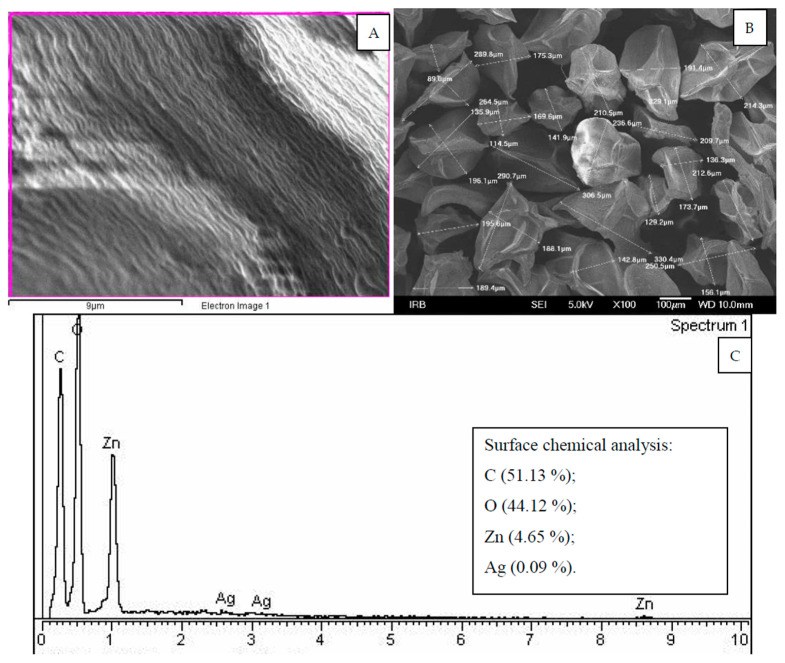
(**A**) SEM microphotograph of the surface of microcapsules, (**B**) determination of the morphological characteristics of the microcapsules, and (**C**) SEM-EDX spectra with the results of investigation of the surface of the microcapsules showing 0.09% of Ag and 4.65% of Zn.

**Figure 6 molecules-27-03219-f006:**
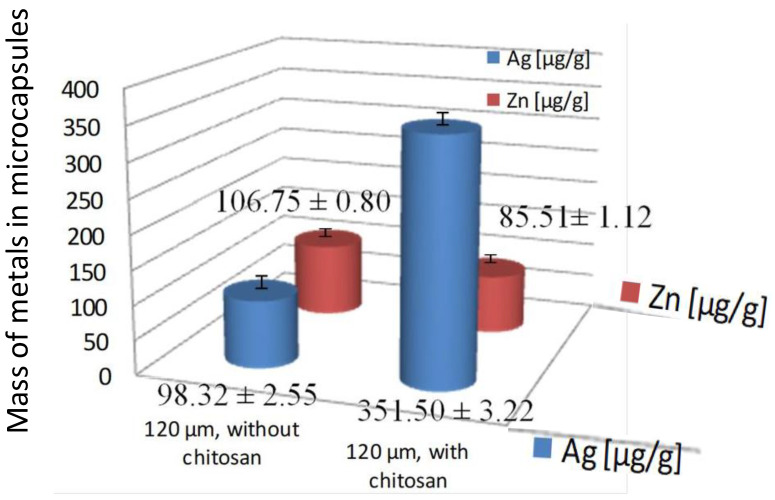
Concentration of silver and zinc in samples of 120 μm microcapsules prepared with and without the chitosan.

**Figure 7 molecules-27-03219-f007:**
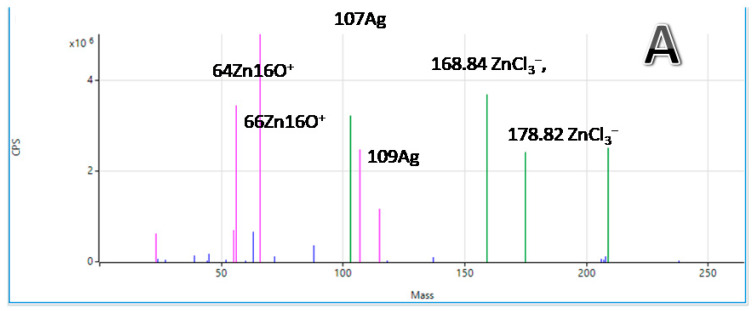
ICP-MS spectra used for determination of the concentration of Ag (107) and Zn (64): (**A**) 120 μm microcapsules prepared without chitosan, (**B**) 450 μm microcapsules prepared without the chitosan, and (**C**) 120 μm microcapsules prepared with the chitosan.

**Table 1 molecules-27-03219-t001:** Analysis of silver and zinc in microcapsules of different diameters (120 or 450 μm) prepared with and without chitosan in their outer shell.

120 μm, without Chitosan		450 μm, without Chitosan		120 μm, with Chitosan	
Element	w [μg/g]	Element	w [μg/g]	Element	w [μg/g]
Ag	98.32 ± 2.55	Ag	190.98 ± 4.87	Ag	351.50 ± 3.22
Zn	106.75 ± 0.80	Zn	121.35 ± 1.65	Zn	85.51 ± 1.12

## Data Availability

Not applicable.

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
