# Peer review of "ICP-MS Determination of Antimicrobial Metals in Microcapsules"

_molecules, 2022, doi:10.3390/molecules27103219_

Round 1

Reviewer 1 Report

The manuscript has been supplemented and expanded, which significantly increases its scientific value. However, in my opinion, the presentation of the results of MS still needs improvement. It would be good if the authors described the individual peaks on MS spectra in Figure 7.

Author Response

Reviewer 1

The manuscript has been supplemented and expanded, which significantly increases its scientific value. However, in my opinion, the presentation of the results of MS still needs improvement. It would be good if the authors described the individual peaks on MS spectra in Figure 7.

Dear Reviewer,

Thank You for your time and for Your valuable comments.

We have made the changes by describing the peaks on MS spectra in the manuscript in two separated parts: firstly in the description of the Figure 7, and secondly in the text of the Discussion part of the manuscript. The changes are marked in green color.

We hope that you will find that the manuscript now much more improved - for which we are very grateful to You.

Reviewer 2 Report

-Authors have respond the comments raised by the reviewers. Now the paper is acceptable in current format. 

-Authors may cite related article recently published in Molecules. 2021, 26(8): 2375. doi: 10.3390/molecules26082375

Author Response

Reviewer 2

-Authors have respond the comments raised by the reviewers. Now the paper is acceptable in current format. 

-Authors may cite related article recently published in Molecules. 2021, 26(8): 2375. doi: 10.3390/molecules26082375

Dear Reviewer,

Thank You for your time and for Your valuable comments.

We have made this change in the Discussion part of the manuscript, as well as in the Literature references of the manuscript as You have suggested, and we hope that you will find that the manuscript is now much more improved - for which we are very grateful to You.

Changes in the manuscript are marked in green color.

This manuscript is a resubmission of an earlier submission. The following is a list of the peer review reports and author responses from that submission.

Round 1

Reviewer 1 Report

Comments:

-English language of the manuscript is poor, should be improved.

-Abstract: ………“The method was applied to determine Ag and Zn in microcapsules samples of different sizes (120 and 13 450 μm) after their preparation with and without chitosan. Significant differences were found in Ag and Zn levels between microcapsules samples prepared with and without chitosan” should be rewritten, and the significance differences values for Ag and Zn should be described.

Introduction: Line 123-125: “Therefore, the goal of this research was to apply the combination of high resolution optical, FTIR spectroscopy and inductively coupled plasma–mass spectroscopy following microwave digestion for detection of metals in different microcapsules filled with silver” there is no correlation between abstract and introduction, authors are required to rewrite both the sections that reflect same information.

-The work is good but the information has not been presented in scientific manners, authors are encouraged to revise the manuscript and resubmit.

Reviewer 2 Report

The paper "ICP-MS determination of metals in microcapsules" is well written and well organized. 

I think it could be ready to be published only after a minor revision of the abstract. In fact, the main goal of the work is missing and not clear in the abstract. 

Reviewer 3 Report

The work presented to me for review is very short and contains only a few experiments. In my opinion, this is not a journal article with such a high IF as Molecules. In addition, editing the manuscript still requires work. It is necessary to edit the captions of figures and the references. When it comes to presenting the results, the figure with the ICP-MS spectrum is missing, the table itself is, in my opinion, insufficient. There is also no discussion of any kind in the paper, the manuscript is simply a presentation of the results of several experiments. I regret to say that it should not be published with Molecules.